# Naphthalene Dehydrogenation on Ni(111) in the Presence of Chemisorbed Oxygen and Nickel Oxide

Kess Marks [1,2], Axel Erbing [2], Lea Hohmann [3], Tzu-En Chien [3], Milad Ghadami Yazdi [1], Matthias Muntwiler [4], Tony Hansson [2], Klas Engvall [3], Dan J. Harding [3], Henrik Öström [2], Michael Odelius [2] and Mats Göthelid [1,*]

[1] Materials and Nano Physics, School of Engineering Sciences, KTH Royal Institute of Technology, SE-11419 Stockholm, Sweden; kessm@kth.se (K.M.)

[2] Department of Physics, AlbaNova University Center, Stockholm University, SE-10691 Stockholm, Sweden; tony.hansson@su.se (T.H.); ostrom@fysik.su.se (H.Ö.); odelius@fysik.su.se (M.O.)

[3] Department of Chemical Engineering, KTH Royal Institute of Technology, SE-10044 Stockholm, Sweden; lhohmann@kth.se (L.H.); chie@kth.se (T.-E.C.); kengvall@kth.se (K.E.); djha@kth.se (D.J.H.)

[4] Photon Science Division, Paul Scherrer Institut, 5232 Villigen PSI, Switzerland

\* Correspondence: gothelid@kth.se

**Abstract:** Catalyst passivation through carbon poisoning is a common and costly problem as it reduces the lifetime and performance of the catalyst. Adding oxygen to the feed stream could reduce poisoning but may also affect the activity negatively. We have studied the dehydrogenation, decomposition, and desorption of naphthalene co-adsorbed with oxygen on Ni(111) by combining temperature-programmed desorption (TPD), sum frequency generation spectroscopy (SFG), photoelectron spectroscopy (PES), and density functional theory (DFT). Chemisorbed oxygen reduces the sticking of naphthalene and shifts $H_2$ production and desorption to higher temperatures by blocking active Ni sites. Oxygen increases the production of CO and reduces carbon residues on the surface. Chemisorbed oxygen is readily removed when naphthalene is decomposed. Oxide passivates the surface and reduces the sticking coefficient. But it also increases the production of CO dramatically and reduces the carbon residues. $Ni_2O_3$ is more active than NiO.

**Keywords:** dehydrogenation; decomposition; naphthalene; nickel; oxygen; nickel oxide





## 1. Introduction

Naphthalene is a prominent component in tar, an unwanted by-product of biomass gasification consisting of a mixture of lighter to heavy hydrocarbons, including a number of polycyclic aromatic hydrocarbons (PAHs). Catalytic conversion of PAHs is a promising method to reduce tar and mitigate their negative effects on gas cleaning and upgrading processes downstream from the biomass gasification process [1–3]. Aside from the destruction of the unwanted tars, converting them into useful permanent gases such as CO and $H_2$ via steam reforming will also increase the total syngas yield of the process. A typical reforming catalyst used under industrial conditions is nickel [4]. Nickel-based catalysts are effective catalysts for the decomposition and dehydrogenation of hydrocarbons but are also prone to poisoning via graphite formation due to carbon formation on the surface [5,6]. The level of carbon poisoning, as well as the ratio of product gases, such as CO and $CO_2$, is heavily influenced by the content of oxygen-containing compounds, such as $H_2O$, in the gas mixture.

Gas-phase water provides surface oxygen in the form of OH and O. Higher content of oxygen-containing compounds therefore leads to more efficient hydrocarbon conversion and oxidation of the carbon deposits [6]. However, a too-high content may lead to oxidation of the nickel catalyst and a subsequent lowered catalytic activity [7].

To study this catalytic system, we use the simplified model catalytic reaction of naphthalene over a Ni(111) single-crystal surface, as naphthalene is one of the most stable compounds in tar and a frequently used model compound. [8]. In a previous study, we have shown that naphthalene adsorbed on a Ni(111) single-crystal surface dehydrogenates, producing $H_2$ gas, as well as graphitic and carbidic surface carbon, rendering the surface inert after several dehydrogenation cycles [9]. In the present paper, we use the same model system to study the effect of oxygen on this decomposition reaction of naphthalene. Whereas in industrial catalytic steam reforming, oxygen is provided via steam in the raw producer gas feed gas [4], in this study, we use varying doses of surface oxygen to simulate the catalytic surface under different oxidative atmospheres from chemisorbed oxygen to oxidation of the outer nickel layer. We use temperature-programmed desorption (TPD), sum frequency generation (SFG), and X-ray photoelectron spectroscopy (XPS) to investigate the surface experimentally. Additionally, density functional theory (DFT) calculations are used to find adsorption geometries and electron-binding energies.

## 2. Results and Discussion

Different starting positions of the naphthalene were tested to probe the set of possible adsorption geometries. The energetically most favorable geometries for naphthalene on Ni(111), on Ni(111)-O (2 × 2), and on NiO are shown in Figure 1a–c. In all three cases, the molecule lies down but with a small tilt.

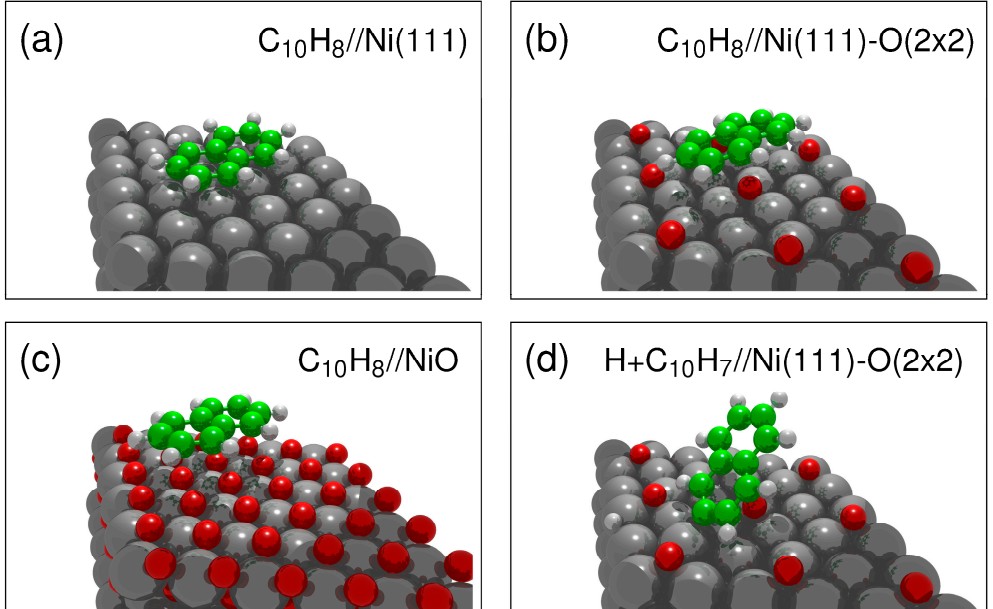

**Figure 1.** The energetically preferred geometries for naphthalene on (**a**) Ni(111), (**b**) Ni(111)-O (2 × 2), (**c**) NiO and (**d**) the first dehydrogenation step for naphthalene on Ni(111)-O (2 × 2).

The adsorption geometry of naphthalene on Ni(111) has been discussed previously [9,10]. The molecule is chemisorbed on the surface, which is indicated by the relatively small distance to the surface [11], approximately 2 Å, close to the sum of the covalent radii of C and Ni [12], as well as by direct inspection of the highest occupied molecular orbitals, which display covalent bonds between the molecule and the surface. Additional details about the adsorption geometries are presented in the Supplementary Material.

In addition, the first dehydrogenation step on the Ni(111)-O (2 × 2) surface is shown in Figure 1d. Previous results from naphthalene dehydrogenation on Ni(111) showed a preference for dehydrogenation of the α-hydrogens (side) [9]. In the case of Ni(111)-O, it is instead the β-hydrogen (end) that is dissociated first. The calculated difference in adsorption energy between α and β dissociation was only 7 kcal/mol [10], and the presence

of surface oxygen changes the balance and instead favors the bonding of the short end to the Ni.

Figure 2 shows TPD spectra from 110 to 700 K of the mass-to-charge ratios 2, 18, 28, 44, and 128 recorded for naphthalene on Ni(111) (a), on $(2 \times 2)$ (b), and on NiO (c).

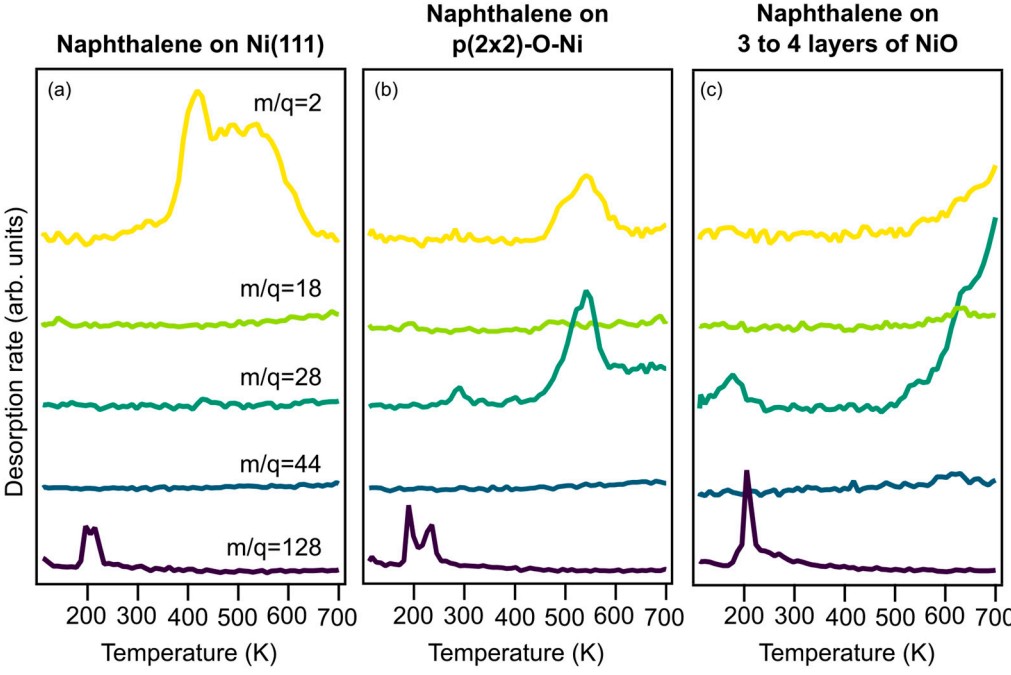

**Figure 2.** TPD spectra for naphthalene on Ni(111) (**a**), on Ni(111)-O $(2 \times 2)$ (**b**), and on NiO (**c**). The traces are shown in order of increasing m/q, starting at the top with 2 ($H_2$) followed by 18 ($H_2O$), 28 (CO), 44 ($CO_2$), and 128 (naphthalene).

Without oxygen, naphthalene dehydrogenation causes a double-peak structure for $H_2$ desorption starting at 380 K and ending at 650 K, as described by Yazdi et al. [9]. The first peak represents the desorption of two $\alpha$-hydrogen atoms from the side of the molecule, where the C-H bonds are replaced by C-Ni bonds together with a tilting of the molecule. Desorption at higher temperatures includes the remaining hydrogens at the same time as a graphene layer is formed on the surface [9].

The only other desorption product from the oxygen-free surface is molecular, physisorbed naphthalene desorbing around 220 K. For all preparations with oxygen, a similar desorption peak of molecular naphthalene (mass 128) is observed, indicating that all preparations produce physisorbed naphthalene. Desorption from the $(2 \times 2)$ surface (Figure 2b) exhibits a $H_2$ desorption peak at 540 K, thus a substantial shift to a higher temperature. A similar decrease in $H_2$ production was previously observed for benzene co-adsorbed with oxygen on Ni(110) [13] and (100) [14]. In addition to $H_2$ desorption at 540 K, CO desorption is observed at the same temperature, indicating that some of the dehydrogenated carbon atoms react with the chemisorbed oxygen and are removed from the surface. This can potentially have a positive effect on the poisoning of the catalyst.

Desorption from NiO decreases hydrogen production even more and moves CO desorption to higher temperatures; the activation energies for naphthalene decomposition are higher on NiO. Although studies of benzene co-adsorbed with oxygen on Pt(111) [15] and Pd(111) [16] show that $H_2O$ and $CO_2$ are the main reaction products, this is not the case for naphthalene co-adsorbed with oxygen on Ni(111). No significant amounts of $H_2O$ (mass 18) and $CO_2$ (mass 44) are observed from any of the surfaces in this study. Only for NiO are small amounts of $CO_2$ and $H_2O$ desorption observed around 600 K. But the main combustion product is CO. This is similar to findings by Viste et al. [17], who found that

CO production dominated over $CO_2$ production with a $CO:CO_2$ ratio above 10, for benzene and oxygen on Rh(111).

Results from the SFG experiments, measuring the C-H stretching region, are presented in Figures 3 and 4. Figure 3 shows spectra at selected temperatures from naphthalene on clean Ni(111), on $(2 \times 2)$, and on NiO. All three preparations show a resonance below 220 K at similar wavenumbers. Numerical fits according to Equation (1) place the resonances at 3052 cm$^{-1}$ for the $(2 \times 2)$ surface and 3057 cm$^{-1}$ for the NiO surface. This matches with results by Marks et al. [10] that show a resonance at 3057 cm$^{-1}$ for naphthalene on clean Ni(111), which was attributed to tilted physisorbed multilayers of naphthalene. Marks et al. observed desorption of the physisorbed naphthalene around 220 K, which seems to be insensitive to oxygen on the surface.

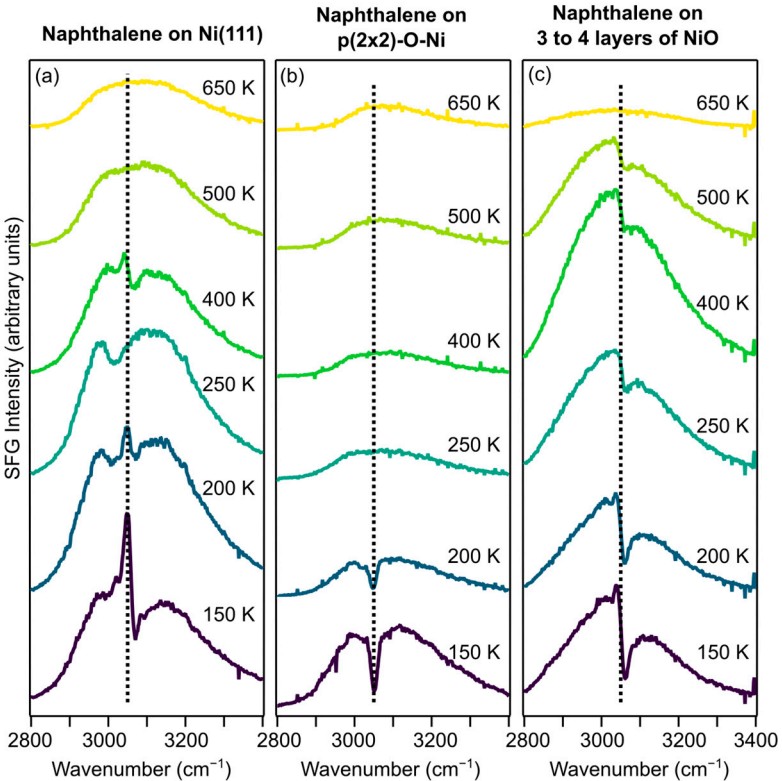

**Figure 3.** Selected SFG spectra at relevant temperatures of the temperature-dependent SFG spectra for the different preparations. The black dotted vertical line is used in each panel to indicate the position of the resonance at 3052 cm$^{-1}$ (**b**) and 3057 cm$^{-1}$ (**a**,**c**).

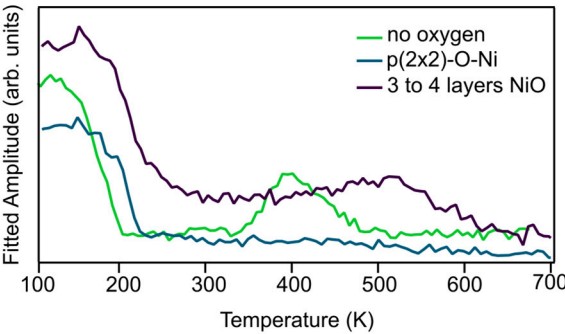

**Figure 4.** Fitted absolute amplitudes of the C-H resonance around 3050 cm$^{-1}$ as a function of temperature, for different surfaces.

The temperature evolution of the SFG amplitudes is plotted in Figure 4. The amplitudes were acquired by fitting each line in the 2D datasets of Figure S1 in the Supplementary

Materials, assuming that the resonance positions (3052 cm$^{-1}$ and 3057 cm$^{-1}$) as well as the other fit parameters are constant, only letting the amplitude vary freely. Around 220 K, the amplitudes decrease for all three surfaces, and for the (2 × 2) surface, the resonance disappears completely. In TPD, we observed desorption of molecular naphthalene (m/q = 128) around 220 K for each preparation, implying that the decrease in the SFG signal around this temperature is due to the desorption of physisorbed naphthalene, similar to what was observed for naphthalene on clean Ni(111) [9].

The SFG signal disappears after the desorption of physisorbed naphthalene from the (2 × 2) surface, as seen in Figure 3. This absence of any resonant signal could imply that no naphthalene molecules remain on the surface. However, since we do observe H$_2$ and CO desorption for these preparations at higher temperatures, it is more likely that chemisorbed naphthalene remains on the surface but is invisible to SFG. There are several reasons why an adsorbate vibration would not be visible in SFG spectroscopy, such as disordered layers and low coverages. Additionally, in SFG, only vibrations that are both Raman and IR active will be observable [18], and according to the metal surface selection rule, only vibrations with a dynamical dipole moment along the surface normal will be active [19]. Thus, a stretching vibration can be invisible if its dynamic dipole moment is parallel to the surface. For naphthalene on Ni(111), C-H stretches are visible in SFG because the nickel repels the hydrogens such that the C-H bonds are tilted with respect to the surface, and thus have a component of their dynamic dipole moment perpendicular to the surface. Chemisorbed oxygen increases the distance and reduces the repulsion of the hydrogens, leaving the C-H bonds essentially parallel to the surface.

For the NiO surface, the resonance at 3057 cm$^{-1}$ decreases but does not completely disappear upon desorption of physisorbed naphthalene. In fact, between 300 and 520 K, the amplitude of this resonance, depicted in red in Figure 4, increases slightly before it decreases and disappears at 620 K. This behavior is different from naphthalene on clean Ni(111), where the chemisorbed naphthalene molecules are strongly de-aromatized[5] and initially give rise to resonance at 2980–3000 cm$^{-1}$ as can be seen in Figure 3a (200–400 K) and is discussed in detail in the paper by Marks et al. [10]. When dehydrogenation starts at 380 K, the partially dehydrogenated naphthalene molecules tilt with respect to the surface and regain aromaticity, which in turn gives rise to a new resonance at 3057 cm$^{-1}$ [10]. The SFG spectra for naphthalene on NiO only have a resonance at 3057 cm$^{-1}$, which is visible from 110 K to 620 K. This implies that naphthalene retains its aromaticity on NiO and adopts a tilted geometry.

For cyclic hydrocarbons, aromaticity is a stabilizing factor and the aromatic resonance persists more than 100 degrees higher on NiO than on Ni(111), implying that naphthalene is more stable on NiO. This reasoning can be turned around with the claim that Ni(111) is a better catalyst for naphthalene decomposition than NiO since the profound de-aromatization upon adsorption weakens the internal bonds of naphthalene, facilitating C-C cleavage at lower temperatures than on NiO. Although co-adsorption of oxygen decreases hydrogen production, it might be beneficial in the long run to add small amounts of oxygen to the Ni(111) catalyst to reduce carbon residues. If the amounts of oxygen are small enough, naphthalene decomposition and hydrogen production can still occur.

Figure 5 shows a C1s spectrum recorded from a monolayer of naphthalene on Ni(111) at room temperature. The numerical fit includes two peaks, one at 284.6 eV and one at 284.1 eV, both with a width of ~0.6 eV. The peak at 284.1 eV represents the eight carbon atoms in the naphthalene molecule with a C-H bond and the peak at 284.6 eV the two carbon atoms with only C-C bonds at the center of the molecule. This assignment is confirmed by DFT calculations, which predict the presence of two peaks 0.5 eV apart. The atomic origins are indicated in the figure. We used the energetically most favorable adsorption geometry, di-bridge- [10], for the C1s calculations. In the experimental spectrum, the ratio is not 1:4 but rather 1:1.4. The additional intensity of the high BE peak is attributed to vibrational excitations. A photoelectron may lose kinetic energy by exciting a C-H vibration on its way from the surface, thus appearing at an apparently higher binding energy in

the spectrum. From SFG, we know that the C-H vibrational energy is around 3000 and 3050 cm$^{-1}$, which corresponds to approximately 0.38 eV. Thus, the energy loss resulting from the C-H vibrational excitation of the low-BE peak will overlap with the high-BE peak.

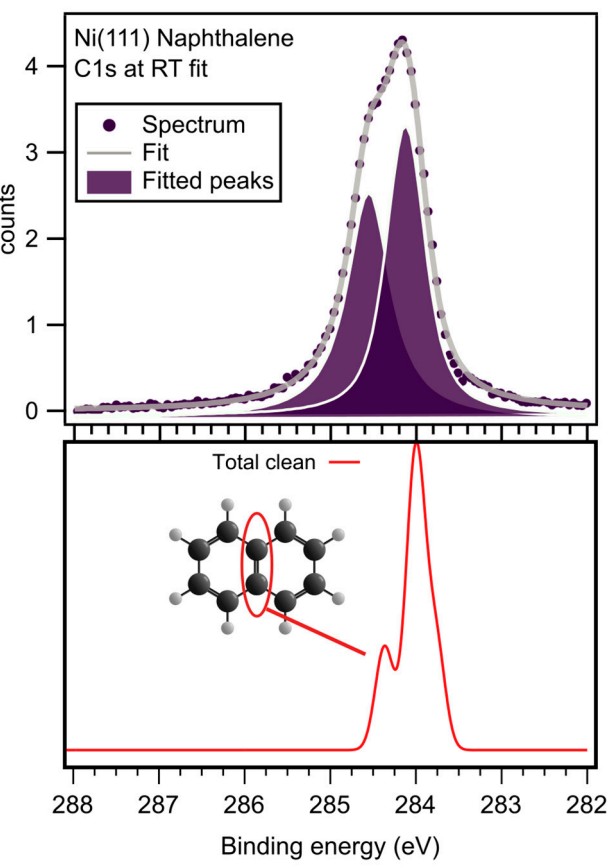

**Figure 5.** Experimental C1s spectrum and numerical fit, recorded from naphthalene monolayer on Ni(111) at room temperature and the corresponding DFT-calculated C1s binding energies in the lower panel.

### 2.1. Adsorption of Naphthalene on Ni/O and NiO

Figure 6 shows C1s and O1s spectra after 10 L naphthalene room temperature exposure on different surfaces. Increasing the dose of oxygen prior to naphthalene leads to decreasing C1s intensity, implying that oxygen (partially) inhibits the adsorption of naphthalene.

Adsorption of naphthalene on the low oxygen doses (from 0 to 3 L) produce C1s spectra that are rather similar. The two C1s components discussed in Figure 5 are present for all low oxygen doses; there is just a small shift of ~0.1 eV to a lower BE between 0 and 3 L of oxygen for both peaks. The relative peak area changes from 1:1.4 at 0 L to 1:2 at 3 L. Note that neither of these ratios is the expected 1:4 based on the molecular structure. As discussed above, this discrepancy is attributed to vibrational excitations. In our SFG discussion, we suggested that naphthalene on (2 × 2) leaves the CH bonds parallel to the surface. The photoelectrons were recorded in normal emission. The cross-section for vibrational excitations is smaller when the electron trajectory is perpendicular to the vibrational direction, so ideally one would expect the ideal 1:4 ratio, but the analyzer has a wide acceptance angle, allowing photoelectrons with a considerable fraction of their momentum along the surface plane and therefore able to excite CH-vibrations. The DFT calculations of the C1s spectrum from naphthalene on (2 × 2) shows a splitting of 0.4 eV between the central and the outer C atoms, which is slightly smaller than for clean Ni(111). We found only a single stable adsorption geometry where one of the central C atoms is positioned on top of a surface O with a surface Ni atom beneath the center of the rings. The molecule is at a slight angle of ~5° and only shows minor bending around its center.

The nature of the adsorption is not as straight forward as for the oxygen free Ni(111) surface, but the relatively large distance from both the surface Ni and O as well as the fact that the calculated spectrum is very similar to the gas phase suggest that the naphthalene is physisorbed in this case. This is also consistent with inspection of the molecular orbitals.

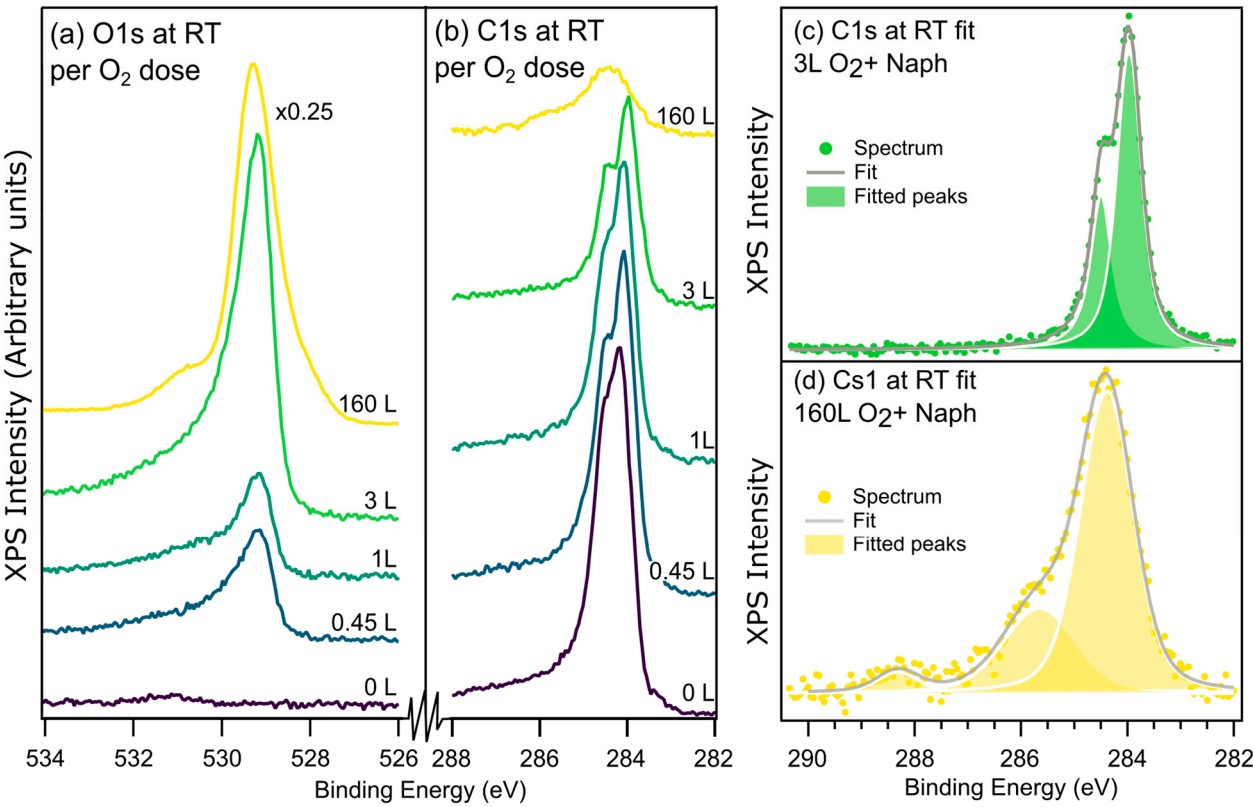

**Figure 6.** O1s (**a**) and C1s (**b**) spectra for a 10 L naphthalene dose on Ni(111) with increasing oxygen doses. Numerical fits of C1s after a 3 L oxygen dose (**c**) and C1s after a 160 L dose (**d**) for comparison. Peak positions in (**c**) are similar to clean nickel (Figure 5), and in (**d**), the peak positions are 288.3 eV, 285.7 eV, and 284.4 eV with widths of 1, 1.4, and 1.2 eV, respectively.

The C1s spectrum for naphthalene on NiO has a different shape as well as a significantly lower intensity (Figure 6a). Peak fitting finds peaks at 288.3 eV, 285.7 eV, and 284.4 eV with widths of 1, 1.4, and 1.2 eV, respectively. The widths are considerably larger than on the metallic Ni(111) surface. This can be explained by a lower conductivity in the oxide and a poorer screening of the core hole and is a common observation when comparing the spectra from metals and their oxides.

The peak at 284.4 eV is likely a convolution of the C-H- and C-C-bonded carbons where the overlap is such that the individual peaks can no longer be resolved. We assign the peaks at 285.7 and 288.3 eV to C-O and C=O (or O-C=O) species, respectively [20]. The presence of these species suggests the (partial) decomposition of naphthalene upon adsorption; this suggestion is strengthened by the presence of OH groups in the O1s spectra at this preparation.

### 2.2. Reaction of Naphthalene on Ni/O (2 × 2) and NiO

Figure 7 shows C1s and O1s spectra from naphthalene adsorbed on Ni(111) (1 × 1), on (2 × 2)-O, and on NiO as a function of temperature. In Figures 8 and 9, we show the numerical fits of selected spectra.

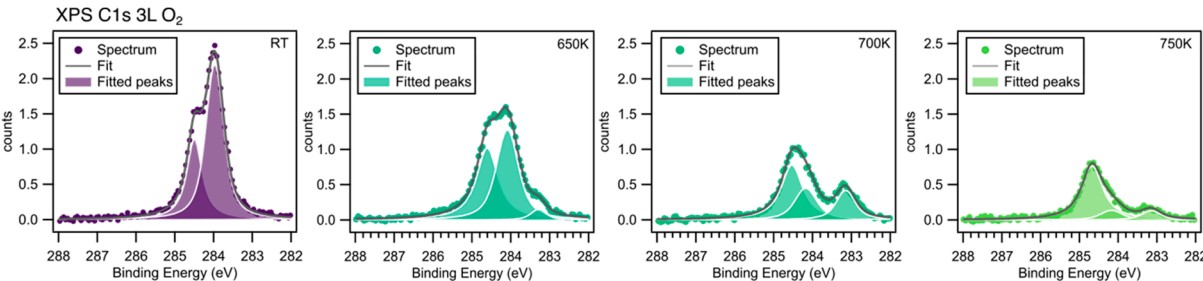

**Figure 7.** Temperature-dependent C1s and O1s spectra recorded after 10 L naphthalene adsorption on Ni(111) (1 × 1) (**a**), Ni(111)-O- (2 × 2) (**b**,**d**), and NiO after 160 L O$_2$ dose (**c**,**e**).

**Figure 8.** Fits of the C1s spectra of the 3 L preparation at four different temperatures, room temperature, 650 K, 700 K, and 750 K.

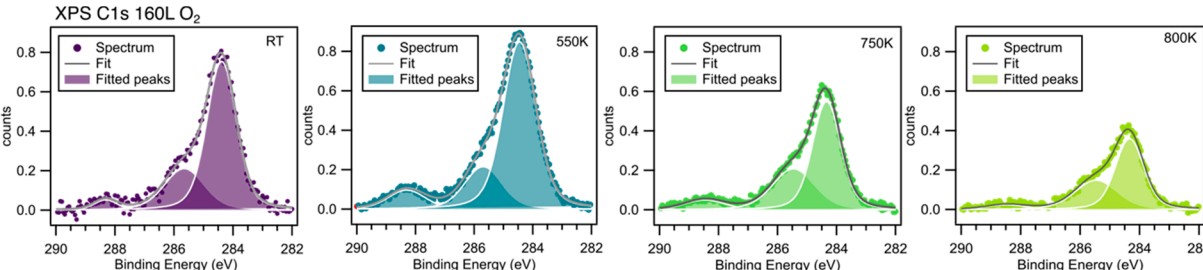

**Figure 9.** Fits of the C1s spectra of the 160 L preparation at four different temperatures, room temperature, 550 K, 750 K, and 850 K.

The left panel of Figure 7 presents spectra recorded from the (1 × 1) surface, going from room temperature (RT) up to 800 K. There are two important observations; around 700 K, there is a new peak appearing at lower binding energy, which was previously assigned to nickel carbide [10]. At increasing temperatures, this peak is gradually removed. Also, the main peak intensity shifts from the C-H to C-C carbon. The shifted intensity thus depicts dehydrogenation and graphene formation.

On (2 × 2), the initial C1s spectrum after RT adsorption is very similar to the spectrum from (1 × 1) at RT. The intensity is lower, indicating a lower sticking coefficient. The effect of annealing is also similar, but the change in the main peak occurs at lower temperatures. These changes happen at the same time as surface oxygen is consumed, as seen in the lower panel.

On NiO, the coverage after the 10 L dose is markedly reduced compared to (1 × 1). The C1s shape is different, but annealing does not affect the line shape until the peak disappears. O1s shows two shifted peaks: one around 529 eV and the other at 531 eV. These are assigned to NiO (529 eV) and $Ni_2O_3$ (531 eV) or defects [21]. Upon annealing, the $Ni_2O_3$ peak is reduced and shifted to higher binding energy. When the C1s signal is zero, $Ni_2O_3$ is consumed. Thus, $Ni_2O_3$ is more active than NiO.

Numerically fitted selected spectra are presented in Figures 8 and 9. Following room temperature adsorption on (2 × 2), the same two C1s peaks are present: C-C at 284.48 and C-H at 283.96 eV. They both have a width of 0.56 eV. At 650 K, these peaks have shifted by ~0.2 eV to 284.61 and 284.17 eV, and the carbidic peak appears at 283.28. Naphthalene decomposition can also be derived from the changes in intensity of the other two peaks. If we compare the fit at RT with the fit at 650 K, we can see that the C-H peak decreases to about 70% of the original intensity, while the C-C peak increases by 20%, indicating that C-H bonds are broken and C-C fragments are created. Not all intensity decreases in the C-H peak can be explained by the growth of C-C and C-Ni peaks; about 15% loss is unaccounted for in the C1s spectra, which is likely due to CO desorption, which is observed to peak at 580 K in the TPD spectra for this preparation.

Heating to 700 K increases and shifts the Ni-C feature to 283.15 eV and decreases the C-H feature even further. At 750 K, the Ni-C and C-H peaks have almost disappeared, whereas the C-C peak is remarkably stable, due to the formation of a stabilizing graphene-like structure. Chemisorbed oxygen adatoms lower the temperature where carbon is removed from the surface. They also facilitate transformation from C-H to C-C.

C1s spectra from naphthalene on NiO at 160 L $O_2$ are shown in Figure 9. Three peaks are required to fit the room temperature spectrum: $COO^-$ at 288.32 eV, CO at 285.66 eV, and the largest one at 284.39 eV representing C-C and C-H with widths of 1, 1.6, and 1.2 eV, respectively. At 550 K, the shape of the spectrum is rather similar, but the intensity of the high-BE (288.3 eV) and low-BE (284.4) peaks has increased. At 750 K, the main peak has significantly decreased in area and the fit places the peaks at 288.48, 285.46, and 284.34 eV with widths of 1.6, 1.7, and 1.2 eV, respectively. At 800 K, the high-BE peak is practically gone and the low-BE peak has shifted to an even lower BE (284.12 eV). This is also the first time we see a significant intensity decrease in the middle peak from 0.35–0.45 in the previous spectra to 0.11 in this last spectrum.

## 3. Materials and Methods

The sample was a Ni(111) single crystal, purchased from the Surface Preparation Laboratory (SPL), Wormerveer, The Netherlands, polished and aligned to within less than 0.1° from the (111) plane. Before each experiment, the sample was cleaned by argon sputtering followed by annealing up to 1050 K in UHV.

The Ni(111) surface was pre-covered with oxygen by dosing $O_2$ gas using a precision leak valve (MDC Precision, Hayward, CA, USA) while keeping the sample temperature at 300 K. Oxygen was added to the surface following the preparation used by [22]: 1–3 L (Langmuir) of $O_2$ was dosed to produce a Ni(111)-O (2 × 2) structure (hereafter called (2 × 2)) with chemisorbed oxygen, >130 L was used to create a thin nickel oxide (NiO) film around 3 layers thick. The cleanness of the surface before dosing as well as the structure of the prepared oxygen surfaces were checked with LEED.

After pre-coverage with oxygen, the surface was exposed to a 10 L dose of naphthalene. At room temperature, this produces a monolayer (ML) [9]. For the SFG and TPD experiments, the sample was cooled to 110 K before the naphthalene dose, whereas in the XPS experiments, the sample was kept at room temperature. This difference in dosing temperature does not affect the monolayer reaction pathway under investigation in the present paper, since the dehydrogenation of naphthalene on Ni(111) starts at about 380 K. Furthermore, multilayers desorb well before room temperature [9,10].

The TPD and SFG measurements were performed in an ultrahigh vacuum (UHV) chamber (Nor-Cal Products Inc., Yreka, CA, USA) with a base pressure of $1 \times 10^{-10}$ Torr. The chamber was equipped with a quadrupole mass spectrometer (QMS, Hiden Analytical, Warrington, UK), LEED instrumentation (SPECS Surface Nano Analysis GmbH, Berlin, Germany), and an ion gun (Omicron Nanotechnology, Taunusstein, Germany) for sample cleaning. The temperature of the sample during the measurements was controlled by a combination of liquid nitrogen cooling and resistive and electron bombardment heating and monitored via a thermocouple that was spot-welded onto the side of the Ni(111) sample.

The TPD measurements were performed by resistively heating the sample at a ramping rate of 50 K/min and recording the desorption rate of various masses continuously using the QMS. The masses chosen included but were not limited to naphthalene, CO, $CO_2$, $H_2O$, $O_2$, and $H_2$. The TPD data were subjected to binning and a linear background subtraction.

For the temperature-dependent SFG experiments, the resistive heating ramping rate was 3 K/min; this is less than for the TPD measurements but necessary to ensure a good signal-to-noise ratio in the SFG data. The laser light used in the SFG experiments was produced by a commercial mode-locked Ti:sapphire amplifier system (Coherent Corp., Saxonburg, PA, USA). The light had a wavelength of 800 nm and an average power of 3.7 W at a repetition rate of 1 kHz, and the pulses had ≤50 fs pulse width. The light from the laser system was split up into two parts; one part was spectrally narrowed to a bandwidth of 12 cm$^{-1}$ using a pulse shaper, and the other part was converted to the mid-IR region using a tunable optical parametric amplifier (TOPAS) and a non-colinear difference frequency generator (n-DFG) (Light conversion, Vilnius, Lithuania). The mid-IR wavelength was set to 3300 nm to cover the C-H stretching region. The narrowed 800 nm and mid-IR beams were sent into the UHV chamber and overlapped spatially and temporally on the sample. The SFG light produced at the sample was sent into a spectrometer (Andor, Oxford Instruments Group, Abingdon, UK) with a grating centered around 640 nm and onto an iCCD camera, where the SFG spectra were recorded with a resolution of 14 cm$^{-1}$.

The experimental spectra were fit assuming that the second-order susceptibility ($\chi^{(2)}$) can be written as

$$\chi^{(2)} = \chi^{(2)}_{NR} + \chi^{(2)}_{R} = \chi^{(2)}_{NR} + \sum_n \frac{A_n e^{i\varphi_n}}{(\omega - \omega_n) + i\Gamma_n} \tag{1}$$

where $\chi^{(2)}{}_{NR}$ is the non-resonant contribution as measured for a clean surface, $\chi^{(2)}{}_R$ is the resonant contribution with $A_n$, $\phi_n$, $\omega_n$, and $\Gamma_n$ being the amplitude, phase, frequency,

and half-width of the resonance. To obtain the temperature evolution of a specific resonance, each slice of the two-dimensional temperature-dependent dataset was fit using the same fitting parameters, only allowing the amplitude to vary freely.

The XPS measurements were performed at the Photo-Emission and Atomic Resolution Laboratory (PEARL) beamline at the Swiss Light Source (SLS), Villigen PSI, Switzerland [23]. The UHV system at PEARL (Omicron Nanotechnology, Taunusstein, Germany) had a base pressure of $3 \times 10^{-10}$ Torr and the preparation chamber was equipped with standard surface science equipment for Ar ion sputtering, annealing, and gas dosing. The analysis chamber at PEARL was equipped with a Scienta EW 4000 analyzer (VG Scienta, Uppsala, Sweden). Photoelectron spectra were recorded for Ni2p, O1s, and C1s using photon energies of 950 eV (Ni2p) 650 eV (O1s), and 380 eV (C1s). The settings were chosen to give a total resolution of 100 meV for C1s, 180 meV for O1s, and 250 meV for Ni2p. The temperature series was recorded by stepwise heating to increasing temperatures and measuring XP spectra after each step. The measured spectra were normalized to the number of scans per measurement as well as the background signal at the high-kinetic energy part of the spectrum. The spectra were fit using Voigt-type peaks in combination with a Shirley background.

## 4. Computational Details

The interaction between naphthalene and oxygen on Ni(111) surfaces was studied theoretically using spin-polarized density functional theory (DFT) calculations under periodic boundary conditions. The initial geometry optimizations and surface relaxations were carried out in the CP2K version 8.0 code [24] using the Gaussian-plane wave method [25,26] and the RPBE and Grimme's D3 van der Waals correction functional [27,28]. The electron density was expanded in a plane-wave basis set with a kinetic energy cutoff of 650 Ry, and the Kohn–Sham orbitals were described using a localized Gaussian DZVP-MOLOPT-GTH-SR 18 e- pseudopotential for Ni and TZVP-MOLOPT-GTH for O, C, and H [29] with GTH pseudopotentials [30,31]. Since the simulated surface is asymmetric, a dipole correction [32] was added to cancel the artificial electric field built up in the slab by the periodic boundary conditions. The geometry optimization was performed with $2 \times 2$ k-point sampling. The total magnetization was fixed in terms of the spin multiplicity corresponding to an average magnetization of 0.61 $\mu_B$ per Ni atom for clean and naphthalene-covered Ni(111) and $(2 \times 2)$ surfaces. For the NiO surface, the total magnetization was chosen assuming that the two top layers of Ni would have a net zero magnetization corresponding to the antiferromagnetic ground state of bulk NiO. The initial atomic magnetic moments of these layers were chosen similarly.

On the optimized adsorbed structures, we performed additional single-point calculations at the Gamma-point, with the addition of a Hubbard correction (DFT+U) to describe the interaction more accurately between the Ni and O for the calculation of adsorption energies and binding energies. A value of 5.7 eV was used as an effective Hubbard U parameter for the Ni top layer of the $(2 \times 2)$ surface and for the two uppermost layers of the NiO surface. In both cases, the lower layers were left unaffected.

The relative C1s and relative O1s binding energies were calculated using the final state approximation [33], also known as the Z + 1 approximation, in which a core electron is removed practically by replacing the excited atom with the Z + 1 element (N and F for C and O, respectively). This substitution results in the removal of a core electron to an empty valence state. Since the wave function of the final state is reoptimized, the generated core hole is screened by the valence electrons. For easy comparison to the experimental XPS spectra, the binding energies were convolved using a Gaussian function. As the Z + 1 approximation does not provide physical total energies and only relative shifts are meaningful to discuss, the calculated binding energies were shifted ad hoc to match the experimental results.



The system unit cell was modeled as a five-layer slab, each layer consisting of $4 \times 4 \times 4$ Ni atoms with two bottom layers kept fixed during the surface relaxation. The $2 \times 2$ geometry was set up with O atoms in fcc sites with a total of 4 Ni atoms per O in the top layer. For NiO, the system was initialized following [34]. Starting from these two surface slabs and the clean Ni(111) slab, the naphthalene molecule was placed on top of the slab, and the system was geometry-optimized.

## 5. Conclusions

We have studied the effect of surface oxygen and surface oxide on Ni(111) during the dehydrogenation and decomposition of naphthalene. We combined temperature-programmed desorption, sum frequency generation spectroscopy, photoelectron spectroscopy, and density functional theory.

Chemisorbed surface oxygen lowers naphthalene adsorption and decomposition activity. The hydrogen production and desorption is shifted to higher temperatures compared to the reaction on clean Ni(111) without the presence of oxygen. CO production is observed through the reaction between naphthalene fragments and surface O. The formation of carbon residues is reduced in the presence of oxygen.

Surface oxide has a serious effect on the decomposition process, a drastically reduced adsorption and a substantial shift to higher desorption temperatures. Both hydrogen and CO start desorbing around 600 K. $Ni_2O_3$ is more active in naphthalene decomposition than NiO and is consumed when naphthalene decomposes and desorbs as CO.

**Supplementary Materials:** The following supporting information can be downloaded at https://www.mdpi.com/article/10.3390/catal14020124/s1, Figure S1: False color plot of temperature dependent SFG spectra recorded from (a) Ni(111) naphthalene, (b) Ni(111) O ($2 \times 2$) naphthalene and (c) Ni(111) NiO naphthalene. The slices presented in figure 3 are cut from the data in this figure; Table S1: Coordinates for optimized structures; Table S2: Naphthalene on Ni(111) O ($2 \times 2$), ground state; Table S3: Naphthalene on NiO; Table S4: Dissociated naphthalene on Ni(111)-O ($2 \times 2$). Flat; Table S5: Dissociated naphthalene on Ni(111)-O ($2 \times 2$). Standing.

**Author Contributions:** Conceptualization, H.Ö., M.O. and M.G.; Investigation, K.M., A.E., L.H., T.-E.C., M.G.Y., T.H., K.E., D.J.H., H.Ö. and M.G.; Resources, M.M., H.Ö. and M.G.; Formal analysis, K.M. and A.E.; Data curation, K.M. and T.-E.C.; Writing—original draft, K.M. and A.E.; Writing—review and editing, A.E., L.H., T.-E.C., M.M., T.H., K.E., D.J.H., H.Ö., M.O. and M.G.; Visualization, K.M. and A.E.; Project administration, K.M., H.Ö., M.O. and M.G.; Funding acquisition, K.E., D.J.H., H.Ö., M.O. and M.G. All authors have read and agreed to the published version of the manuscript.

**Funding:** This research was funded by Swedish research council (VR), Stiftelsen Olle Engkvist Byggmästare (SOEB), and the Swedish Foundation for Strategic Research SSF (ITM17-0236). The work was carried out within the Swedish Gasification Center (SFC) consortium. Funding from the Swedish Energy Agency and academic and industrial partners is gratefully acknowledged. The Paul Scherrer institute is kindly acknowledged for providing beamtime at the PEARL beamline. The computations were enabled by resources provided by the National Academic Infrastructure for Supercomputing in Sweden (NAISS) and the Swedish National Infrastructure for Computing (SNIC) at NSC and PDC partially funded by the Swedish Research Council through grant agreements no. 2022-06725 and no. 2018-05973.

**Data Availability Statement:** Data are contained within the article and Supplementary Materials.

**Conflicts of Interest:** The authors declare no conflict of interest.

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
