# Peer review of "Naphthalene Dehydrogenation on Ni(111) in the Presence of Chemisorbed Oxygen and Nickel Oxide"

_catalysts, doi:10.3390/catal14020124_

Round 1

Reviewer 1 Report

Comments and Suggestions for Authors

This is a very interesting and important study for investigating the catalytic pyrolysis and gasification of biomass. But it needs some revisions before accepting:

1) How to control the layers of the adsorbed naphthalene? Can you show the results of the spectral analysis?  How to inhibit the molecular island formation during adsorption?  

2) How about the real results about the catalyst passivation basing on this study? 

3)There are many mistakes in English spelling and punctuation. Please check them carefully and correct them.

Author Response

Comments and Suggestions for Authors

This is a very interesting and important study for investigating the catalytic pyrolysis and gasification of biomass. But it needs some revisions before accepting:

1) How to control the layers of the adsorbed naphthalene? Can you show the results of the spectral analysis?  How to inhibit the molecular island formation during adsorption?  

At room temperature adsorption stops at one monolayer. That has been described in our previous work and is cited in the current paper. The spectral analysis is presented in figures 8 and 9. Island formation in the form of mulitayer islands in inhibited by the high vapour pressure of naphthalene at room temperature.

2) How about the real results about the catalyst passivation basing on this study? 

The introduction has been revised and proper references added.

3) There are many mistakes in English spelling and punctuation. Please check them carefully and correct them.

We have carefully checked the manuscript. One of us is English and fluent in his native language.

Reviewer 2 Report

Comments and Suggestions for Authors

In the manuscript, the authors studied on the effect of presence of chemisorbed oxygen and nickel oxide on the dehydrogenation of naphthalene over Ni(111) surface based various spectroscopy technologies and DFT methods. They found a increase in the stability of the catalysts with pre-adsorbed oxygen species but sacrifice the activity. The study is very comprehensive and their conclusions is based on solid experiments. However, there are some small issues that need to be addressed before its publications.

1, The introduction is very short lacking of a very sound motivation of the study. By adding previous or updated related studies and novelty of this work will make it much more complete.

2, In the computational details, the reference for PBE functional is not correctly cited, the authors should check and cite the right one.

3, The gamma point is chosen for the calculations barely based on the size of the slab doesn't scientifically make sense. A rigorous convergence test should be conducted. Or a more intensive kpoints should be used.

4, I'm not fully convinced by the authors' method on setting the initial magnetic moment of the system, especially for NiO system.

5, All the optimized structure should be provided or at least the coordinates should be attached to SI.

6, For the DFT calculations results, it will be much better to provide the full results of the adsorption energy on all studied system. And for the decomposition of naphthalene , it will make much more sense to calculate the reaction barrier and make comparations on both the binding strength and the barrier.

7, Is that a typo in the second paragraph on page 6 which is highlighted in bold black?

Author Response

Reviewer 2

1, The introduction is very short lacking of a very sound motivation of the study. By adding previous or updated related studies and novelty of this work will make it much more complete.

We have revised the introduction, and believe that the study is now well motivated.

2, In the computational details, the reference for PBE functional is not correctly cited, the authors should check and cite the right one.

We apologize for the wrong statement of functional, now corrected to RPBE, and the faulty references in the theory section. For some reason all references from the theory and thereafter seems to be shifted.

3, The gamma point is chosen for the calculations barely based on the size of the slab doesn't scientifically make sense. A rigorous convergence test should be conducted. Or a more intensive kpoints should be used

K-point sampling 2x2 was indeed used for the geometry optimization, but in CP2K if was not possible for the DFT+U calculations which are limited to the Gamma point. The text is now corrected.

4, I'm not fully convinced by the authors' method on setting the initial magnetic moment of the system, especially for NiO system.

This has been expalined and described in the revised version. We believe that our previous description was a bit unclear

5, All the optimized structure should be provided or at least the coordinates should be attached to SI.

All coordinates are now attached in the supporting information.

6, For the DFT calculations results, it will be much better to provide the full results of the adsorption energy on all studied system. And for the decomposition of naphthalene , it will make much more sense to calculate the reaction barrier and make comparations on both the binding strength and the barrier.

This would be very intersting yes. There is a new student working on this and will expand the computational work further. We believe that this represents a substantial additional work that will require time. Further, the current paper is essentially experimental with theoretical support. The future paper is mainly computational with new additional supporting experiments.

7, Is that a typo in the second paragraph on page 6 which is highlighted in bold black?

Yes it was a typo and has now been corrected.

Round 2

Reviewer 1 Report

Comments and Suggestions for Authors

The manuscript was well revised, and it can be acceptable for publication.